# Synergistic insights into human health from aptamer- and antibody-based proteomic profiling

Maik Pietzner [1,2], Eleanor Wheeler[1], Julia Carrasco-Zanini [1], Nicola D. Kerrison[1], Erin Oerton[1],
Mine Koprulu [1], Jian'an Luan [1], Aroon D. Hingorani [3,4,5], Steve A. Williams [6],
Nicholas J. Wareham [1,5] & Claudia Langenberg [1,2,5✉]

Affinity-based proteomics has enabled scalable quantification of thousands of protein targets in blood enhancing biomarker discovery, understanding of disease mechanisms, and genetic evaluation of drug targets in humans through protein quantitative trait loci (pQTLs). Here, we integrate two partly complementary techniques—the aptamer-based SomaScan® v4 assay and the antibody-based Olink assays—to systematically assess phenotypic consequences of hundreds of pQTLs discovered for 871 protein targets across both platforms. We create a genetically anchored cross-platform proteome-phenome network comprising 547 protein–phenotype connections, 36.3% of which were only seen with one of the two platforms suggesting that both techniques capture distinct aspects of protein biology. We further highlight discordance of genetically predicted effect directions between assays, such as for PILRA and Alzheimer's disease. Our results showcase the synergistic nature of these technologies to better understand and identify disease mechanisms and provide a benchmark for future cross-platform discoveries.

[1] MRC Epidemiology Unit, University of Cambridge, Cambridge, UK. [2] Computational Medicine, Berlin Institute of Health (BIH) at Charité – Universitätsmedizin Berlin, Berlin, Germany. [3] Institute of Cardiovascular Science, Faculty of Population Health, University College London, London WC1E 6BT, UK. [4] UCL BHF Research Accelerator Centre, London, UK. [5] Health Data Research UK, London, UK. [6] SomaLogic, Inc, Boulder, CO, USA. ✉email: claudia.langenberg@mrc-epid.cam.ac.uk

Proteins are the essential functional units of human metabolism that translate genomic information and enable growth, development and homeostasis. Naturally occurring sequence variation in the human genome, either in close physical proximity to the protein-encoding gene (*cis*) or anywhere else in the genome (*trans*), has wide-ranging effects on proteins, including, but not limited to, expression, structure and function, with important implications for human health[1,2]. Early studies have started to describe the genetic architecture of protein targets measured in plasma but all have been small-scale or restricted to one platform[3–9].

Modulating protein abundances or function represents the most common mode of action of drugs[10] and major pharmaceutical companies now integrate protein quantitative trait loci (pQTLs) into their strategies to identify new drug targets or to repurpose existing drugs[11–13]. This has only been possible through the commercial development and application of scalable affinity-based proteomic techniques that can measure thousands of protein targets simultaneously. Projects are now underway to apply these techniques to large-scale studies, such as the UK Biobank[14,15], which will provide major scientific opportunities. However, information about the consistency of protein measures and the pQTLs identified using different proteomics platforms is needed to inform the generalisability of genetic findings and strategies for future data integration or meta-analytical approaches, and, more importantly, for possible downstream consequence for biomedical applications, for example, the alignment of pQTLs with disease-causing genetic variants.

Here we assess 871 proteins targeted by two complementary techniques, the SomaScan v4 assay[16] (aptamer-based) and Olink's proximity extension assay[17] (PEA, antibody-based), measured in up to 10,708 individuals, including overlapping measurements by both technologies in a subset of 485 participants. We use a machine learning approach to identify technical parameters and protein characteristics that contribute to measurement variation between platforms. We identify hundreds of pQTLs and systematically assess their consistency in a reciprocal design, generating a unique benchmark for future studies. We create a comprehensive, genetically anchored cross-platform protein-phenotype network using colocalisation analysis at protein-encoding loci across thousands of phenotypes, identifying substantial synergy between both platforms. We show that protein–phenotype colocalisation seen with only one platform goes beyond missing target specificity and can be explained by alternative proteoforms induced by genetic variants altering the amino acid sequence of the protein and the effects of alternative splicing.

## Results

We used the SomaScan v4 platform (SomaLogic Inc., Boulder, Colorado, US) to measure protein abundances of 4775 unique human protein targets (covered by 4979 unique aptamers) from frozen EDTA-plasma samples in 12,345 participants from the Fenland study[18] (Supplementary Data 1). We assessed 1069 protein targets based on 1104 measures across 12 Olink® Target 96-plex panels, based on the PEA technology using the same EDTA-plasma samples from 485 Fenland study participants. Measurements were performed by the manufacturers and methods have previously been described in detail[19,20] and are provided in the Methods section. We identified a total of 871 overlapping proteins targeted by 937 unique SomaScan–Olink reagent pairings (Fig. 1, see Methods).

**Technical factors affecting correlations between protein targets**. We observed varying correlation coefficients for overlapping

measurements with a median of 0.38 (IQR: 0.08–0.64) spanning almost the entire range from high positive (Leptin, $r = 0.95$) to inverse correlations (Heat shock protein beta-1, $r = -0.48$) (Fig. 2a and Supplementary Data 2). When we used the SomaLogic data without a normalisation step applied to correct for unwanted technical variation and to make data comparable across cohorts, we observed a higher median correlation (median: 0.50, IQR: 0.19–0.72) (Fig. 2a). While correlation coefficients seemed to increase, we observed substantial differences in the association with various phenotypic characteristics comparing normalised to non-normalised data (Supplementary Fig. 1). For example, systolic blood pressure was associated with 3745 aptamers in the entire SomaScan data using the non-normalised compared to 1528 in the normalised data set. Such an effect might be explained by phenotypic variation that is associated with median fluorescence intensities across proteins per sample, which can introduce artificial associations.

We identified technical factors, such as binding affinity of the SOMAmer reagent or missing/extreme measurements (likely due to technical variation and strong genetic effects, see Supplementary Note 1), and certain protein characteristics, for example, presence of a transmembrane domain, to explain varying correlation coefficients, based on a random-forest-based feature selection algorithm (Fig. 2b, see Methods and Supplementary Note 1). In line with previous findings[21], protein targets that have been validated by orthogonal methods, such as mass spectrometry-based target validation or immunoassays, showed higher correlation coefficients as well (median correlation: 0.57 vs 0.27, $p$ value $< 1.59 \times 10^{-21}$). These results were largely independent of the normalisation procedure used and we considered the normalised SomaScan data as the primary resource in the following analyses.

**Shared genetic effects are target-dependent**. We identified a total of 1923 genetic variant–SOMAmer–Olink triplets with evidence from at least one platform (816 SOMAmer reagents, 770 Olink measures and 1267 genetic variants, Supplementary Fig. 2 and Supplementary Data 3, see Methods) and observed considerably lower correlation coefficients between effect estimates (Fig. 3) than previously reported[5], with values of 0.41 for *cis*-pQTLs and 0.34 for *trans*-pQTLs. Correlation coefficients were higher (*cis*-pQTL: 0.68, *trans*-pQTL: 0.75) for well-correlating protein targets (Supplementary Fig. 3) and comparable to an independent set of Olink-based pQTLs[8] (Supplementary Fig. 4 and Supplementary Data 4).

We next tested more rigorously for a shared genetic architecture across platforms and identified 306 (63.9%) genomic region-to-protein target associations that were shared between platforms, that is, showed the same, directionally consistent genetic signal (see Methods and Supplementary Fig. 5 and Supplementary Data 5), with approximately similar fractions for *cis*- and *trans*-pQTLs out of 479 with sufficient power for replication (Fig. 3b). This included 13 regions for which we discovered two independent *cis*-pQTLs ($r^2 < 0.1$) for SomaScan but only the secondary signal was shared with Olink (Supplementary Figs. 6 and 7). The remaining 36.1% genomic region–protein target associations were platform-specific because they were either (1) only evident for one of the two assays (24.6%, $N = 59$ for the SomaScan assay and $N = 59$ for Olink), or (2) showed evidence for distinct genetic signals at the same locus (10%, 48 pairs). We observed a lower fraction of shared genomic regions when comparing to publicly available Olink pQTLs[8] with 39.1%, which was best explained by the presence of multiple non-specific *trans*-pQTLs (see Supplementary Note 1 and Supplementary Data 6).

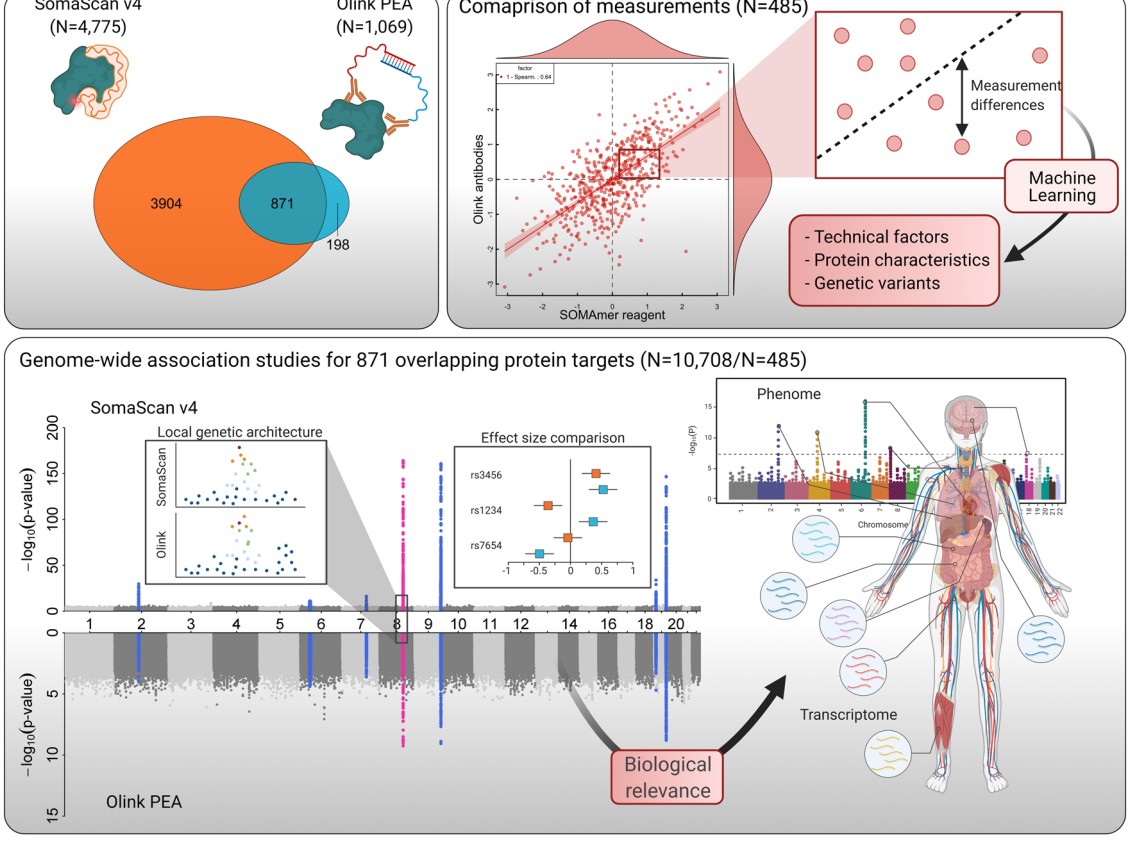

**Fig. 1 Scheme of the study design.** The Venn diagram displays the overlap in protein targets captured by the SomaScan assay and the Olink proximity extension assay (PEA). Modes of binding to the protein target are depicted simplified next to each ellipse. Correlation coefficients were used to compare both technologies and factors possibly accounting for measurement differences and low correlation coefficients examined in a subset of 485 individuals with overlapping measurements. For the set of 871 common protein targets, genome-wide association analysis was performed in 10,708 (SomaScan assay) and 485 (Olink PEA) participants in the Fenland cohort. Correspondence of genetic associations was analysed by examining local genetic architecture, comparison of effect estimates and evaluation of phenotypic consequences. Parts of this figure have been created with BioRender.com.

Based on this assessment we identified the following factors to be associated with a higher likelihood of a distinct or platform-specific pQTL: (1) a lower observational correlation, (2) lower binding affinity of the SOMAmer reagent to the protein target, (3) linkage to a protein altering variant (PAV) (in particular for *cis*-pQTLs discovered using the SOMAscan assay), (4) a high proportion of extreme values in SomaScan measurements and (5) missing colocalisation with *cis*-expression QTLs (eQTLs) and phenotypic traits (Fig. 3c and Supplementary Data 7–9), by evaluating meta-regression models (see Methods).

Finally, we observed that genotypes significantly affected the correlations of measurements between platforms. We identified 22 instances in which the correlation coefficient between measurements of the same protein target across both platforms significantly differed by genotype (false discovery rate <20% for an interaction term), including pQTLs in *cis* and *trans* (Fig. 3d). In other words, the agreement between both platforms was higher for a genetically defined subgroup of participants, with effects in *cis* possibly pointing to epitope effects, whereas effects in *trans* pointing towards posttranslational modifications, such as glycosylation (Supplementary Data 10 and Supplementary Note 1). For example, we identified that stratifying the correlation of YKL-40 ($r = 0.45$ overall) by rs2071579 improved up to 0.96 among carriers of the minor C-allele. rs2071579 is in almost complete LD ($r^2 = 0.99$) with the missense variant rs880633 (p.R145G), the major C-allele (AF = 53% in Fenland) introduces an arginine to glycine substitution in a predicted antibody binding sequence of YKL-40, thereby likely attenuating the binding capacity of the

aptamer reagent. As this results in a constant dilution factor depending on the genotype, correlations between the affected SomaScan assay and the possibly unaffected Olink assay improve upon stratification.

**A genetically anchored protein–phenotype network across platforms.** We created a gene-protein–phenotype network, to systematically explore the synergy of cross-platform proteomic studies to identify and better understand disease mechanisms. To this end, we performed phenome-wide colocalisation screens for all 871 protein-encoding regions mapping to the set of overlapping protein targets using region-wide summary statistics derived from both proteomic platforms (Fig. 4, see Methods). We identified shared genetic signals for a total of 547 protein target–phenotype pairs (posterior probability >80%), comprising 112 protein targets and 342 phenotypes (Supplementary Data 11). About a third (36.3%) of the gene-protein–phenotype pairs were only seen for one of the two platforms, including 108 pairs unique to Olink and 91 pairs unique to SomaScan accounting for the differences in statistical power. A few (1.4%) showed strong evidence for a shared signal with proteins measured by both platforms but with opposing effect directions. We further identified four pairs that were consistent across platforms once the effect of the lead *cis*-pQTL for the SomaScan assay has been taken into account, indicating recovery of biological plausible information by accounting for possible measurement artefacts. Finally, about a third (34.3%)

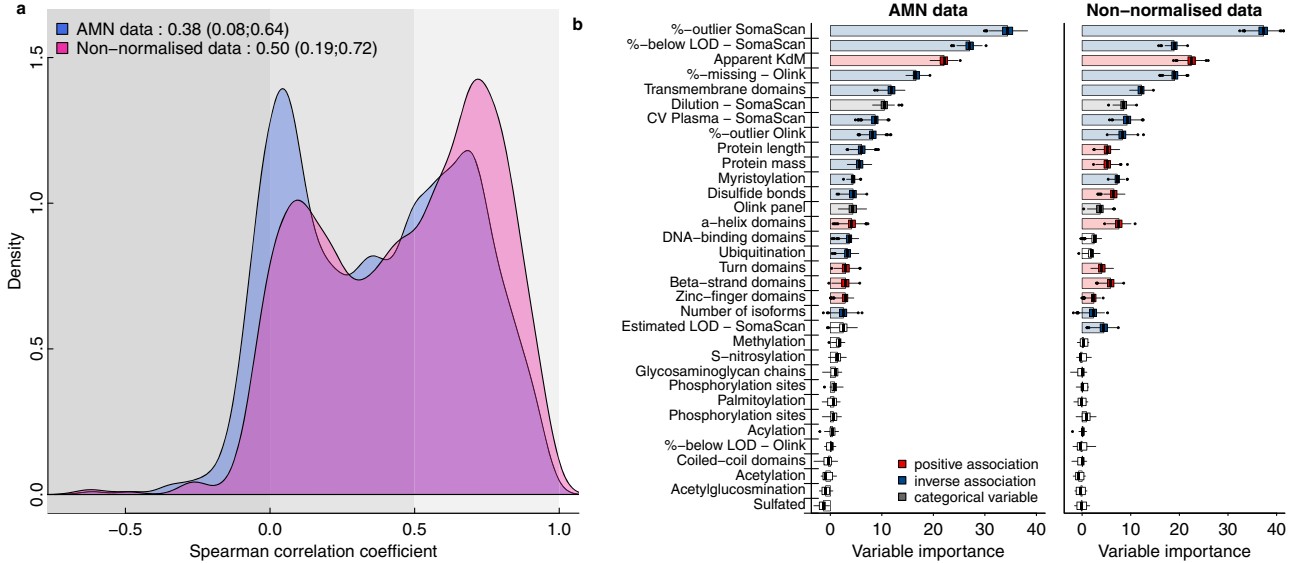

**Fig. 2 Summary of correlations between measurements on both platforms. a** Distribution of correlation coefficients across 937 mapping aptamer–Olink measure pairs ($n = 871$ unique protein targets). Source data can be found in Supplementary Data 2. **b** Importance measures derived from a random-forest-based variable selection procedure to predict Spearman correlation coefficients across all 871 protein targets, including technical factors and protein characteristics. Coloured boxplots indicate variables for which the importance measure remained significant after accounting for multiple testing (adjusted $p < 0.01$). Boxplots display the distribution of importance measures for each variable across 500 bootstrap samples. For the purpose of visualisation, median values have been highlighted by bars. % below LOD = fraction of measurement values below the detection limit of the assay. Source data are provided as a Source Data file.

of the gene–protein–phenotype pairs were consistent between both platforms.

**Directionally discordant associations at an Alzheimer's disease locus.** We identified eight protein target–phenotype pairs for which proteins as measured by both assays were highly likely to share the same genetic signal with the same phenotype but with opposite effect directions for the same protein target or its isoforms (Figs. 4 and 5a). For instance, the missense variant rs1859788 (p.G78R, sAF = 31.7% for the A-allele) in *PILRA* was the lead *cis*-pQTL inversely associated with paired immunoglobulin-like type 2 receptor alpha (PILRA) measured by Olink (beta = −0.74, $p < 3.48 \times 10^{-29}$). In contrast, we found positive associations for the same genetic signal with two SOMAmer reagents targeting soluble isoforms of the same protein (6402-8 targeting isoform FDF03-deltaTM (beta = 1.26, $p < 2.67 \times 10^{-5193}$) and 10816-150 targeting isoform FDF03-M14 (beta = 1.26, $p < 1.53 \times 10^{-5360}$)), but no association with the SOMAmer reagent designed to target the canonical protein (8825-4, beta = 0.004, $p = 0.75$). Statistical colocalisation provided strong evidence of a genetic signal shared between all three different protein measures and Alzheimer's disease (Supplementary Data 11 and Fig. 5a), in line with the A-allele of rs1859788 having been identified as protective for Alzheimer's disease[22]. PILRA is an inhibitory receptor expressed in dendritic and myeloid cells[23] and p.G78R was shown to reduce signalling via reduced ligand binding, likely modulating microglia migration and activation in the brain[22]. G78R is located in the extracellular-domain common to all three forms of PILRA[23]. Therefore, the positive effect directions of the SOMAmer reagents targeting the two isoforms in the absence of an association with the canonical protein suggest aptamer binding affinity introduced by p.G78R being restricted to the soluble isoform. However, our results cannot distinguish which isoform the polyclonal Olink antibodies target and whether the inverse association reflects reduced binding affinity to the variant protein of at least some of them.

We identified similar examples with possible downstream consequences for phenotypic interpretation, including hepatoma-derived growth factor and high-density lipoprotein cholesterol concentrations or intracellular adhesion molecule 1 and lymphocyte cell count (Supplementary Data 11).

**A phenotypically distinct role of cis-pQTLs for IL-7 receptor subunit alpha.** We observed a segregation of phenotypes colocalising at the *IL7R* locus depending on the protein assay used to identify *cis*-pQTLs for the IL-7 receptor subunit alpha (IL-7Ra) (Fig. 4 and Supplementary Fig. 8). The lead *cis*-pQTL rs6451229 (MAF = 40.1%) for the SomaScan assay colocalised with type 1 diabetes and treatment for hypothyroidism, whereas the lead *cis*-pQTL, rs11742270 (MAF = 26.8%), for the Olink assay colocalised with multiple sclerosis, allergic disease, primary biliary cirrhosis and basophil counts (Supplementary Data 11). Both variants are only in weak LD ($r^2 = 0.25$) and the phenotypic divergence further supports two distinct signals. The lead variant for Olink is in perfect LD with a well-known splice variant (rs6897932, $r^2 = 1$) previously shown to mediate increased risk for multiple sclerosis by skipping of exon 6 and creating a higher amount of soluble IL-7Ra[24] and has since been identified for various immune-related diseases[25]. A higher fraction of soluble, that is, circulating, IL-7Ra might explain the positive association of the same genetic variant with IL-7Ra as measured by Olink. Soluble IL-7Ra has been suggested as active IL-7 reservoir, including an increased risk for the generation of IL-7-dependent self-reactive T cells in autoimmunity[25]. With respect to the lead *cis*-pQTL for SomaScan, our finding supports *IL7R* as a likely causal gene at an established type 1 diabetes locus[26]. More recent work, however, identified two distinct variants (rs2303137[27] and rs2287900[28], $r^2 = 0.29$) in the same locus, both in moderate LD ($r^2 = 0.45$) with the SomaScan *cis*-pQTL but without evidence for colocalisation. However, there is some orthogonal evidence supporting *ILR7* as the candidate causal gene at this locus, including the preliminary success of IL-7Ra antibodies in mouse models of type 1 diabetes[29] and immunomodulation in patients with type 1 diabetes[30].

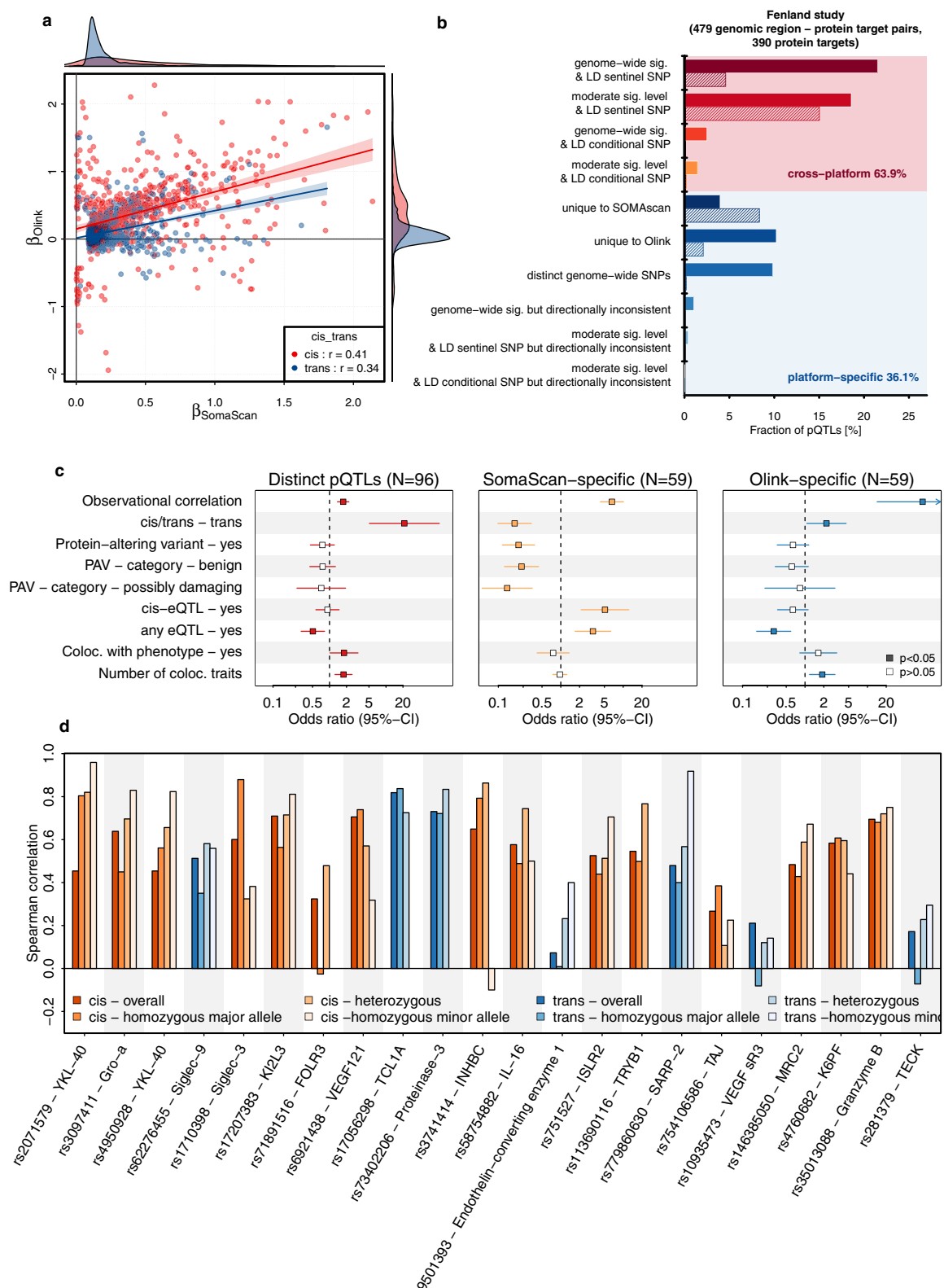

**Allelic heterogeneity at FGCR2A and autoimmune disease**. We identified three independent genetic variants at the *FCGR2A/FCGR2B* locus that acted in a platform- and phenotype-specific manner on the gene products low-affinity immunoglobulin gamma Fc region receptor II-a (FCGR2A) and II-b (FCGR2B) as measured by SomaScan and Olink (Fig. 4 and Supplementary Fig. 9). We identified rs7515174 (G-allele, allele frequency = 11.2%) as an

intronic *cis*-pQTL unique to FCGR2A measured by Olink (beta = −1.25, p value < 7.5 × 10⁻⁴¹) and a shared signal with rheumatoid arthritis (RA) in Europeans[31] (beta = −0.11, PP > 84.9%). The variant is in strong LD (r² = 0.99) with the multi-nucleotide variants rs9427397 and rs9427398, which cumulatively introduce a substitution of glutamine with tryptophan (p.Q63W) at position 63 of the protein sequence (based on transcript ENST00000271450.6) and

**Fig. 3 Consistency of genetic effects across platforms. a** Comparison of beta estimates from linear regression models across 816 corresponding SOMAmer–Olink pairs ($n = 770$ unique protein targets) with at least one genome-wide associated genetic variant for either of the two, including 1267 distinct genetic variants ($R^2 < 0.8$). Colouring is based on the genomic location of genetic variants. Red indicates variants close to the protein-encoding gene (cis, ±500 kb) and blue otherwise. Estimates are presented in Supplementary Data 3. **b** Summary of platform agreement for 479 genomic region–protein target associations with sufficient power among the Fenland subsample with available Olink measures ($N = 485$). More information is detailed in Supplementary Data 5. **c** Factors associated with pQTLs that are shared across platforms compared to three sets of platform-specific controls. Odds ratios and 95% confidence intervals for factors associated with cross-platform protein quantitative trait loci (pQTL) across the SomaScan v4 and Olink assays (Supplementary Data 9). The panels are based on 540 variant–protein target pairs (306 shared, 234 platform-specific) with sufficient power for replication in the Fenland sample. PAV protein altering variant, eQTL expression quantitative trait loci, Coloc. colocalisation, GWAS genome-wide association analysis. **d** Spearman correlation coefficients stratified by genotype. The first bar in each column indicates the overall correlation, and the three successive bars indicate the correlation among homozygous carriers of the major allele, heterozygous carriers and homozygous carriers of the minor allele (if any). Colours indicate whether the pQTL was in cis (orange) or trans (blue). Protein target–pQTL pairs were selected based on a linear regression model (see main text). Source data are provided as a Source Data file.

---

thereby possibly reducing the affinity of the Olink antibodies for the modified protein.

A different, independent variant ($r^2 < 0.14$), rs4657041 (MAF = 48.0%), was the lead intronic cis-pQTL for FCGR2A measured by SomaScan as well as FCGR2B measured by Olink. This signal was shared with ulcerative colitis (UC, PP > 95.3%), systemic lupus erythematosus (SLE, PP > 82.6%) and various cell surface markers of different immune cell populations, including FCGR2A (CD32) (Fig. 4). FCGR2A is an activating receptor upon binding of immunoglobulin (IgG) complexes as part of the humoral immune system and rs4657041 is in strong LD ($r^2 = 0.99$) with the missense variant rs1801274 encoding for the well-known low-/high-responder phenotype[32,33]. The substitution of histidine for arginine conferred by the A-allele at position 134 increases binding of IgG2, a mechanism suggested to contribute to a higher risk for autoimmune disease, including UC and SLE. GWAS studies, however, showed opposing effect directions for UC (beta = 0.14, $p$ value < $1.5 \times 10^{-18}$) and SLE (beta = $-0.18$, $p$ value < $5.5 \times 10^{-11}$). The extremely strong effect of rs1801274 on the SomaScan measure of FCGR2A (>1 s.d. per A-allele, beta = $-1.21$, $p$ value < $1.1 \times 10^{-6276}$) likely provides a simple proteomic readout for low versus high-responder status relevant for immunotherapy using antibodies[34]. We note, that possibly both cis-pQTLs for FCGR2A relate to RA, since we obtained evidence that the lead signal for SomaScan colocalised with RA (PP > 87.1%) as assessed in Biobank Japan (Supplementary Data 11). Finally, rs6665610 (a synonymous variant within FCGR2B) was a cis-pQTL unique to FCGR2B as measured by SomaScan with no evidence for a shared phenotypic association. Together, these results suggest that SomaScan and Olink target different forms of FCGR2A, each with distinct downstream consequences for human health as evidenced by the colocalising genetic signal.

**Phenotypic colocalisation unique to the SomaScan assay.** Aptamers of the SomaScan assay are designed to bind through their specific shape to the target protein. This shape-based nature enabled us to discover multiple unique protein–phenotype links, including cathepsin H and type 1 diabetes (rs2289702 within CTSH), TREM-like transcript 2 protein and monocyte count (rs62396355 within TREML2) or plexin-B2 and systolic blood pressure (rs28379706 within PLXNB2) (Supplementary Data 11).

We identified a complete segregation of abundance- versus shape-based effects for growth-differentiation factor 15 (GDF-15). GDF-15 is generally considered as a stress signal inducing weight loss and reducing food intake via an effect on aversion to food, a phenomenon thought to explain cachexia/anorexia in

cancer patients and episodes of hyperemesis during pregnancy[35]. We observed that the SomaScan-specific cis-pQTL (rs75347775, MAF = 24.5%) showed strong evidence for colocalisation with related phenotypes, including a self-reported measure of childhood obesity and coffee intake, and was further in strong LD ($r^2 = 0.96$) with a known risk variant (rs45543339) for hyperemesis gravidarum[36]. The lead cis-pQTL for GDF-15 as measured by Olink and replicated in a larger study[8] (rs1227734, MAF = 14.0%), however, was unrelated to these outcomes, albeit being a secondary signal for GDF-15 as measured by SomaScan (Fig. 5b). The lack of association between rs75347775 and the Olink measure likely indicates that the genetic variant acts via a differential recognition by the SOMAmer reagent and rs75347775 is indeed in strong LD with the missense variant p.H202D (rs1058587, $r^2 = 0.98$, Supplementary Fig. 10). The G-allele mediating the amino acid substitution was associated with higher GDF-15 recognition by the SomaScan assay (beta = 0.39, $p < 4.7 \times 10^{-174}$), but with 32% reduced risk for hyperemesis gravidarum (odds ratio: 0.68, 95% CI: 0.62–0.75, $p < 3.4 \times 10^{-14}$) as well as a higher risk for childhood obesity (beta = 0.01, $p$ value < $6.7 \times 10^{-13}$) and reported coffee intake (beta = 0.01, $p$ value = $5.6 \times 10^{-8}$) clearly opposing the well-documented effects of high circulating GDF-15[35].

GDF-15 is being actively investigated as an anti-obesity agent[37]. However, instrumenting genetic variants, including the ones found with the Olink assay, that are proxies for life-long higher GDF-15 levels in the physiological range did not provide evidence for a causal role of GDF-15 in measures of adult obesity and metabolic health[8,38]. We, however, obtained evidence that the same missense variant underlying childhood obesity colocalises with adult body mass index (PP = 95.1%, beta = 0.01, $p$ value < $8.2 \times 10^{-8}$) once stronger independent signals in the region have been taken into account (Fig. 5b). Our findings therefore provide human genetic evidence that it is not the abundance of GDF-15 within the physiological range but rather the proteoform (p.H202D or p.H6D in the mature protein) that possibly has an effect on food aversion, an effect of particular relevance during childhood, in which food choices are more restricted compared to later life.

**Phenotypic colocalisation unique to the Olink assay.** We report 34 cis-pQTLs unique to the Olink assay and identified in only 485 samples that showed strong evidence for colocalisation with phenotypic traits (PP > 80%) (Supplementary Data 11). These included established cardiovascular risk loci such as FGF5 (e.g., hypertension[39], coronary heart disease[40] or atrial fibrillation[41]) and UMOD (e.g., hypertension and estimated glomerular filtration rate[42]) for which we estimate that genetically higher protein

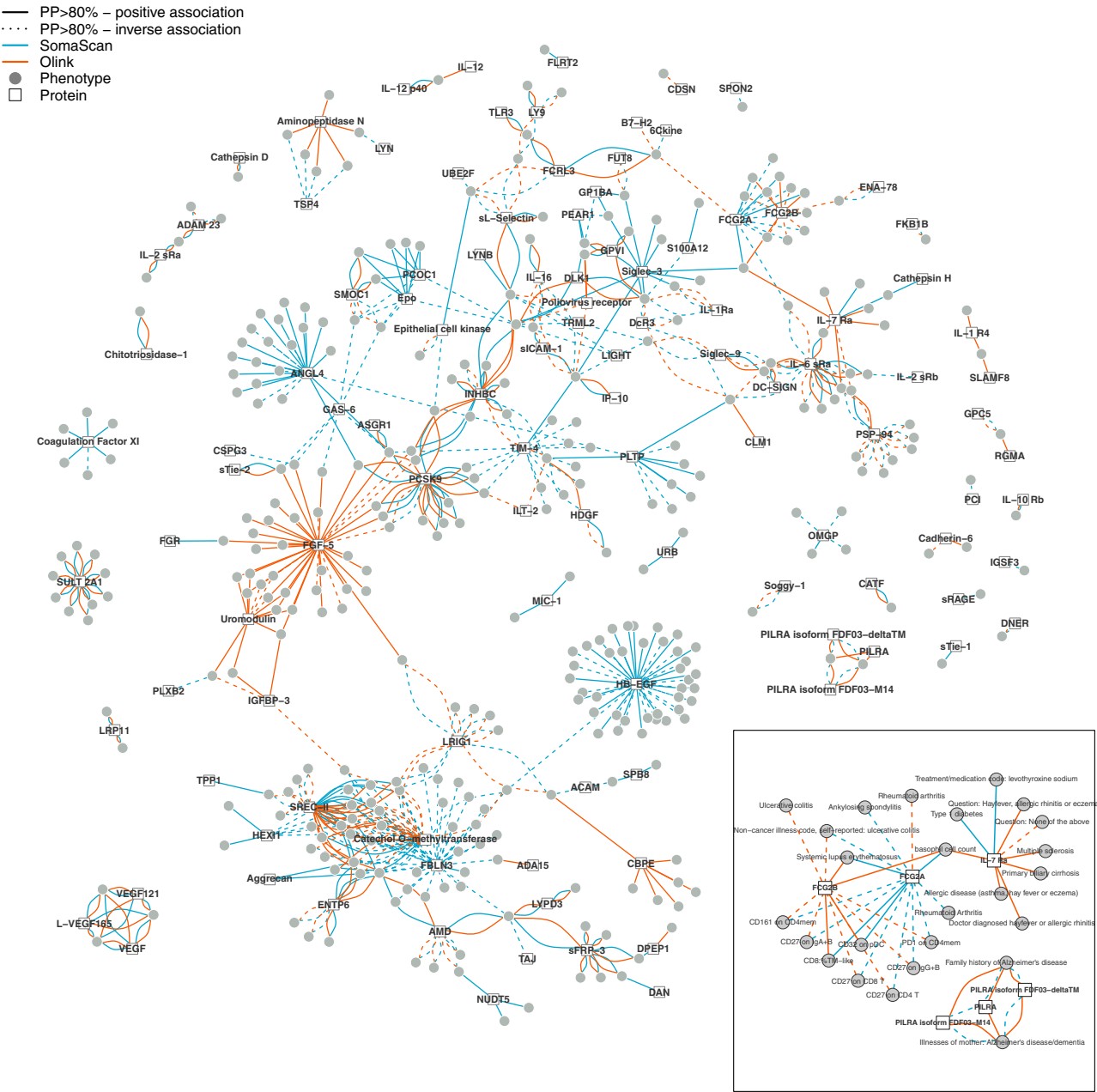

**Fig. 4 A genetically anchored protein–phenotype network.** Each node is either a protein target (square) or a phenotype (circle) and a connection was drawn between both if the protein target shared a genetic signal (posterior probability (PP) >80%) with the phenotype within a ±500 kb region around the protein-encoding gene (Supplementary Data 11). Colours indicate whether the shared signal was identified using the SomaScan (cyan) or the Olink assay (orange). Protein–phenotype connections consistent between both platforms are indicated by two lines connecting the protein and the phenotype. Solid lines indicate a positive association of the shared genetic variant with the phenotype aligned to the protein-increasing allele. The inset highlights selected subnetworks for which both proteomic techniques provide complementary information.

levels are causally associated with higher disease risk, for instance, a 1 s.d. increase in genetically predicted FGF-5 levels was associated with a 12% higher risk of coronary artery disease (odds ratio: 1.12; 95% CI: 1.08–1.16; p value < $9.0 \times 10^{-12}$) possibly via its effect on hypertension (1.32; 1.29–1.35; p value < $1.7 \times 10^{-99}$). Other known or recently described disease loci for which we identified evidence for a shared gene-protein–phenotype signal included carboxypeptidase E (*CPE*) and bone mineral density[43], ICOS ligand (*ICOSLG*) and RA[31], or SLAM Family Member 8 (*SLAMF8*) and Crohn's disease[44] pointing towards biomarkers of disease progression or probably druggable targets such as for Aminopeptidase N *(ANPEP)* and eye morphology[45].

## Discussion

Identification of DNA sequence variants modulating protein levels or activities and shared with disease loci can identify disease-causing mechanisms and help to prioritise new and repurpose existing drug targets[11]. To inform and advance such strategies, comparison across different measurement techniques can not only validate identified signals, but also help to better understand the potential biological relevance of platform-specific signals for human health. We provide genetically anchored evidence that the integration of diverse proteomic techniques enables the identification of disease mechanisms beyond changes in the abundance of circulating proteins, emphasising the need

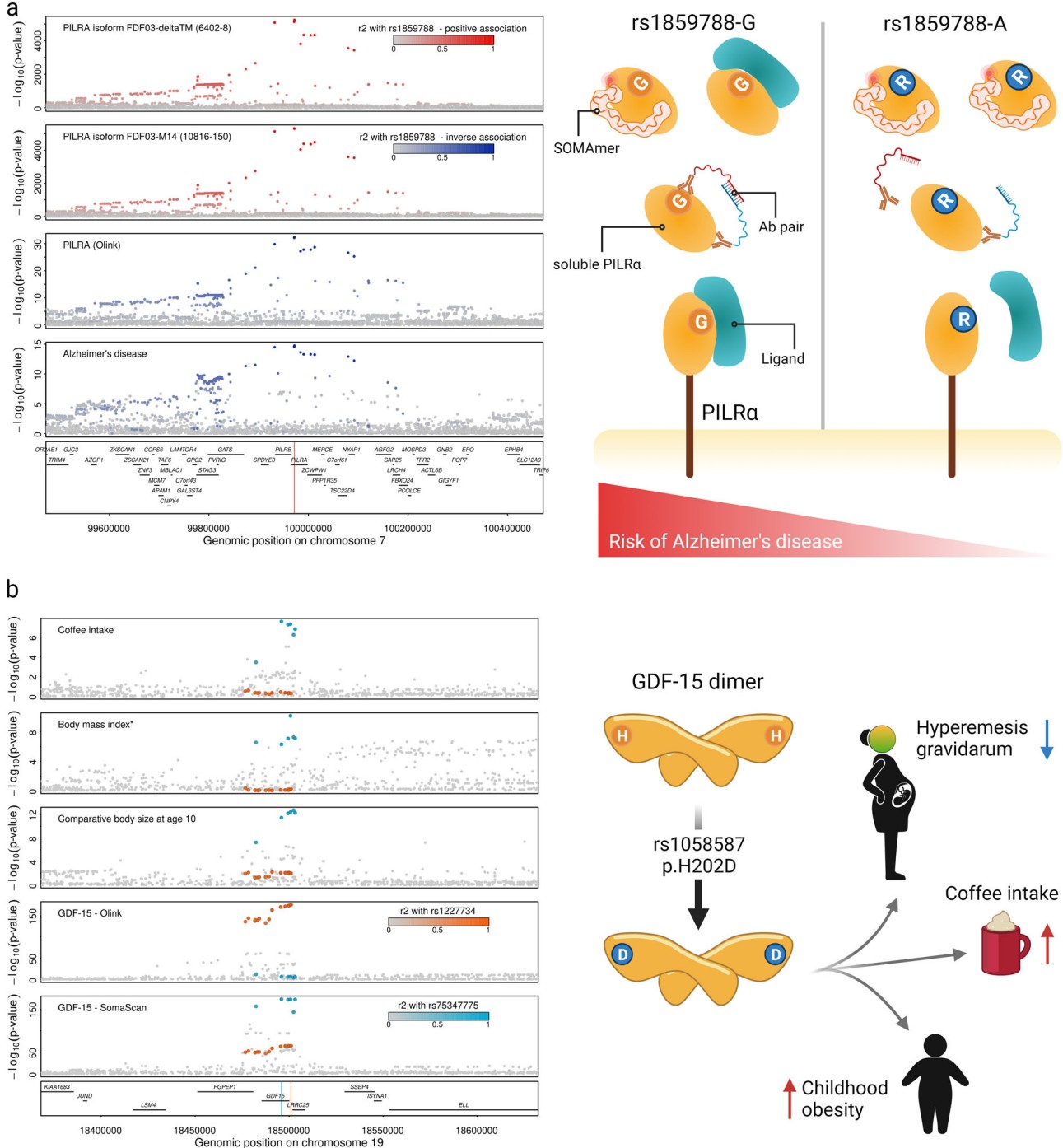

**Fig. 5 Regional association plots for the *PILRA* and *GDF15* locus. a** Regional association plots for paired immunoglobulin-like type 2 receptor alpha (PILRα) measured by SomaScan (top rows) and Olink, as well as for Alzheimer's disease centred around a colocalising signal for the missense variant rs1859788 within *PILRA* (p.G78R). Colours indicate direction of effect for the A-allele of rs1859788 on the respective trait (blue—inverse, red—positive) and shading indicates linkage disequilibrium ($r^2$) with the lead variant at the locus. The red line in the gene panel indicates the position of the variant. *P* values for protein measures were derived from genome-wide association analysis (linear regression models) from the Fenland cohort as described in the main text, whereas summary statistics for Alzheimer's disease was obtained from Jansen et al.[58]. The scheme on the right displays a possible mechanism by which the p.G78R could lead to discordant results between SomaLogic and Olink. **b** Each panel shows summary statistics (*p* values) from genetic association studies for coffee intake, comparative body size at age 10, body mass index (conditioned on lead signals), and growth-differentiation factor 15 (GDF-15) measured by Olink and SomaScan. The lead genetic variants for each assay as well as variants in high linkage disequilibrium are highlighted by colours (blue—SomaScan, orange—Olink). Summary statistics for phenotypes were obtained from the Open GWAS database (UK Biobank)[55] and protein summary statistics for GDF-15 from[8] for Olink and SomaScan from the present study. The scheme on the right shows possible consequences of a differently shaped GDF-15 protein. A 3D model of the alternative protein is presented in Supplementary Fig. 10. Parts of this figure have been created with BioRender.com. Source data are provided as a Source Data file.

for complimentary techniques and most importantly better understanding of the relevance of platform-specific pQTLs for protein function.

A common pattern among results not shared between both assays related to the reliance of the SomaScan assay to a conserved protein structure to enable aptamer binding (Supplementary Data 10). This has important implications for protein-level based causal inference techniques, such as Mendelian randomisation, where genetic instruments acting in *cis* are commonly used to infer plasma 'abundance' rather than function of the encoded protein. Biased conclusions from such techniques could arise where the direction of the protein binding affinity ('abundance') and the function of the mutant protein the variant is instrumenting are disconnected. Employing intermediate traits, that is, those that lie on a causal pathway from the protein to the disease, instead to obtain genetic weights for such analysis may help to address this problem. We further show that strong and platform-specific signals with extreme binding affinity can mask signals that are shared across platforms and demonstrate that association statistics conditioning on such strong lead pQTLs can uncover biologically relevant signals shared between platforms.

A common theme of platform-specific *cis*-pQTLs that aligned with the genetic signal for phenotypic consequences was a genetically induced alternative form of the target protein, which we referred to as 'proteoform', such as for GDF-15 for which we obtained evidence that an alternative form of the protein rather than altered abundances may mediate downstream effects. While this generally pointed towards specificity of the affinity reagent to the 'canonical' protein (or at least the protein sequence that has been used to select the affinity reagent against with) and cannot be distinguished from as a technical artefact, triangulating genetic variation with protein 'abundance' (or presence) and phenotypic consequences provided evidence for the candidate causal gene and the expression of the alternative proteoform at substantial levels to be detected in plasma. More importantly, such effects enabled us to derive functional hypothesis that go beyond reduced or enhanced expression of certain protein targets starting to explore functional proteomics in humans.

Previous smaller scale studies[3,5,21] have performed unidirectional validation of pQTLs for a selected set of protein targets and reported inflated correlation estimates due to missing alignment of effect directions to the protein-increasing or -decreasing allele, thereby introducing an artificially large reference range. We provide a systematic and bidirectional characterisation of pQTLs covering 871 overlapping protein targets and show that the majority of pQTLs are shared across platforms (64%) but with substantially lower correlations than previously reported in *cis* and *trans*. We identify factors associated with platform-specific pQTLs for both platforms, which can directly help to inform strategies for prioritising pQTLs in academic and pharmaceutical efforts that have used either platform at scale, in particular for the thousands of protein targets only assayed by the Somalogic platform. Unlike our previous effort demonstrating the feasibility of meta-analysing genetic signals for metabolites measured by diverse platforms[46], the proteome possesses distinct challenges and requires tailored strategies to increase samples size by integrating diverse platforms. Our results provide a benchmark and guidance for any future genetic studies aiming to increase samples size by integrating proteomic data across different platforms.

We identify several characteristics affecting the correlation between both assays, including technical variation, certain protein characteristics and a strong effect of genetic variants (Fig. 6). However, the lack of full technical details of the assays that are not in the public domain as they are commercially sensitive and general methodological differences between the assays did not permit a more rigorous assessment of non-biological factors. This includes the similarity of synthetic peptides used to select binding reagents or a measure of binding affinity for antibodies, which might likely yield additional insights into possible differences. Incorporation of complementary techniques such as mass spectrometry may help to resolve some of these issues[47], for example by linking a pQTL to an actually measured peptide sequence, which would provide important scientific opportunities if the approach can be applied at scale. In addition, structural characterisation of proteins bound to affinity reagents using mass spectrometry has the potential to identify the concrete protein species bound to the affinity reagent[4,21]. While we identify factors that increase the likelihood of cross-platform pQTLs, larger studies are needed to test for factors differentially associated with replication of *cis*- and *trans*-pQTLs.

By integrating strong evidence for gene-protein–outcome pairs across two complementary proteomic techniques, we were able to identify hundreds of examples (>30% of all), which would have otherwise been missed using only one technique. While both techniques have their merits, mutual application in clinical and population-based studies, possibly further complemented with mass spectrometry, is unfeasible but using genetics as a common anchor across studies along with well-powered GWAS for phenotypes enables novel discoveries for individual diseases and among diseases as exemplified in the protein–phenotype network.

## Methods

**MRC Fenland cohort**. The Fenland study is a population-based cohort study of 12,435 participants, predominantly of White British ancestry born between 1950 and 1975. Participants were recruited from general practice surgeries in the Cambridgeshire region of the UK and underwent detailed phenotyping at a baseline visit between 2005 and 2015 (Supplementary Data 1). Exclusion criteria were clinically diagnosed diabetes mellitus, inability to walk unaided, terminal illness (life expectancy of ≤1 year at the time of recruitment), clinically diagnosed psychotic disorder, pregnancy or lactation. The study was approved by the Cambridge Local Research Ethics Committee (NRES Committee – East of England Cambridge Central, ref. 04/Q0108/19) and all participants provided written informed consent. The consent covered measurements made from blood samples as well as extends beyond the baseline examination. As previously described[18], participants in the study were on average 48.6 years old (standard deviation: 7.5 years) and 53.4% were female.

**Proteomic measurements**. Relative protein abundances were measured in fasting EDTA-plasma samples from 12,084 Fenland Study participants collected at the baseline visit by SomaLogic Inc. (Boulder, US) using an aptamer-based technology (SomaScan V4 assay). Details of the assay have been described previously[20], but briefly, 4775 human protein targets were evaluated by 4979 aptamers; short single-stranded DNA molecules, which are chemically modified to specifically bind to protein targets and quantified using DNA microarrays. To account for variation in hybridisation within runs, hybridisation control probes are used to generate a hybridisation scale factor for each sample. To control for total signal differences between samples due to variation in overall protein concentration or technical factors such as reagent concentration, pipetting or assay timing we applied adaptive median normalisation. Briefly, a ratio between each aptamer's measured value and a reference value, derived from healthy external controls (Covance data set, described in Williams et al.[20]) is computed, and the median of these ratios is computed for each of the three dilution sets (20%, 0.5% and 0.005%) and applied to each dilution set to centre the study medians to the reference medians. The study set is then normalised by scaling each protein signal by the respective scale factors. Samples were removed if they were deemed by SomaLogic to have failed or did not meet our acceptance criteria of 0.25–4 for all scaling factors. In addition to passing SomaLogic QC, only human protein targets were taken forward for subsequent analysis (4979 out of the 5284 aptamers). Aptamers' target annotation and mapping to UniProt accession numbers as well as Entrez gene identifiers were provided by SomaLogic.

We estimated a limit of detection for each SOMAmer reagent using a 'robust estimate' method suggested by SomaLogic, based on the median plus 4.9 × median absolute deviation (MAD) signal of the blank (buffer) samples. We further defined outliers for SOMAmer and Olink measurements as being outside the median ±5 × MAD based on test sample signals and used the fraction of outliers as a variable to explain variation.

Plasma samples for a subset of 500 Fenland participants were additionally measured using 12 Olink 92-protein panels using PEAs[17]. Of the 1104 Olink proteins, 1069 were unique (*n* = 35 on >1 panel, average correlation coefficient 0.90). We imputed values below the detection limit of the assay using raw

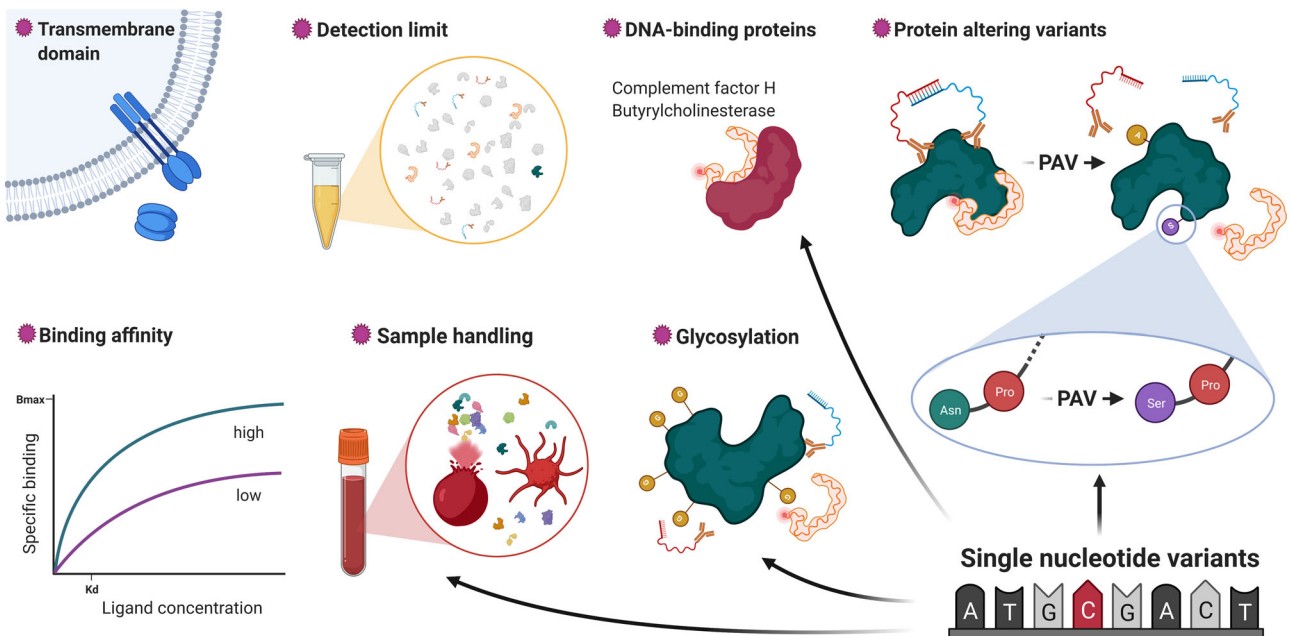

**Fig. 6 Sources of variation.** Graphical summary of factors contributing to variation in the affinity-based discovery of the plasma proteome. PAV protein altering variant, SNV single-nucleotide variant. This figure has been created with BioRender.com.

fluorescence values. Protein levels were normalised ('NPX') and subsequently log2-transformed for statistical analysis. A total of 15 samples were excluded based on quality thresholds recommended by Olink, leaving 485 samples for analysis. Participants were selected at random from the largest set of Fenland participants that had been examined at the same test site and were genotyped on the same array to minimise technical artefacts for the platform comparison. Demographics of the subcohort were identical to the overall cohort (Supplementary Data 1).

**Protein target mapping**. We identified overlapping protein targets between both techniques using either UniProt identifiers (www.uniprot.org) or based on the same encoding gene as provided by the manufacturers. Where multiple measurements were available for a protein assayed on multiple Olink panels, we selected one of the protein measures from one of the panels at random for two reasons. Firstly, Olink uses the same type of antibodies irrespective of the panel and secondly, the average correlation was 0.90 (range 0.68–0.99) for the same protein target across different panels. We kept each SOMAmer reagent matching to one Olink reagent for downstream analysis, since they bind to distinct structural characteristics of the protein target[16]. This procedure yielded 937 unique SOMAmer–Olink measurement pairs, comprising 871 unique protein targets (Fig. 1 and Supplementary Data 1). We further queried the UniProt database to obtain protein domain information and other characteristics of overlapping protein targets. We noted that protein targets overlapping between platforms were enriched for secreted proteins (odds ratio: 3.66, $p$ value < 4.7e−44) and high-affinity targets (odds ratio: 1.18, $p$ value < 4.3e−6), and slightly depleted for protein targets with a higher amount of outlying samples (odds ratio: 0.87, $p$ value < 1.3e−4) when comparing to the entire set of proteins captured by the SomaScan v4 assay.

**Statistical analysis**. We used rank-based inverse normal transformations to make protein measurements between both technologies comparable and reported Spearman rank-based and Pearson correlation coefficients as a measure of concordance between platforms.

To derive factors explaining the Spearman correlation gradient across protein targets, we created a matrix with meta-information for each protein target, including information about technical characteristics of each platform as well as characteristics of the protein target (Fig. 2) and used those as input for a Random-forest-based feature selection approach, called Boruta-feature selection[48]. Briefly, this method employs multiple rounds of Random-forest generation and includes so-called shadow variables, which are permuted versions of the original input variables, to derive test statistics for the variable importance measure.

**Genotyping and imputation**. Fenland participants were genotyped using one of three genotyping arrays: the Affymetrix UK Biobank Axiom array (OMICS, $N = 8994$), Affymetrix SNP5.0 (GWAS, $N = 1402$) and Illumina Infinium Core-Exome 24v1 (Core-Exome, $N = 1060$). Samples were excluded for the following reasons: (1) failed channel contrast (DishQC <0.82); (2) low call rate (<95%); (3) gender mismatch between reported and genetic sex; (4) heterozygosity outlier; (5) unusually high number of singleton genotypes or (6) impossible identity-by-

descent values. Single-nucleotide polymorphisms (SNPs) were removed if: (1) call rate <95%; (2) clusters failed Affymetrix SNPolisher standard tests and thresholds; (3) MAF was significantly affected by plate; (4) SNP was a duplicate based on chromosome, position and alleles (selecting the best probeset according to Affymetrix SNPolisher); (5) Hardy–Weinberg equilibrium $p < 10^{-6}$; (6) did not match the reference or (7) MAF = 0.

Imputation to the HRC (r1) panel for the autosomes of the OMICS and GWAS subsets was performed using IMPUTE4[49] and to HRC.r1.1 for the Core-Exome subset and the X-chromosome (all subsets) using the Sanger imputation server[50]. Imputation to the UK10K+1000Gphase3[51] panel using the Sanger imputation server was used to supplement the HRC imputation with additional variants not present in that panel. We excluded variants with MAF < 0.001, imputation quality (info) <0.4 or Hardy–Weinberg Equilibrium $p < 10^{-7}$ in any of the genotyping subsets from further analyses.

**GWAS and meta-analysis**. After excluding ancestry outliers and related individuals, up to 10,708 Fenland participants ($n = 485$ for Olink proteins) had both phenotypes and genetic data for the GWAS (OMICS = 8350, Core-Exome = 1026, GWAS = 1332). We transformed aptamer abundances to follow a normal distribution using the rank-based inverse normal transformation and then adjusted for age, sex, sample collection site and ten principal components in STATA v14. Residuals from the regression were used as input for the genetic association analyses. Test site was omitted for protein abundances measured by Olink as those were all selected from the same test site. Genome-wide association was performed under an additive model using BGENIE (v1.3)[49] and we combined the results for the three genotyping arrays using a fixed-effects meta-analysis in METAL[52]. Following the meta-analysis, 17,652,797 genetic variants also present in the largest subset of the Fenland data (Fenland-OMICS) were taken forward for further analysis.

For each protein target, we used a genome-wide significance threshold of $1.004 \times 10^{-11}$ (SomaScan) or $4.5 \times 10^{-11}$ (Olink) and defined non-overlapping regions by merging overlapping or adjoining 1 Mb intervals around all genome-wide significant variants (500 kb either side), treating the extended MHC region (chr6: 25.5–34.0 Mb) as one region. We classified pQTLs as *cis*-acting instruments if the variant was less than 500 kb away from the gene body of the protein-encoding gene.

We performed conditional analysis as implemented in the GCTA software using the *slct* option for each genomic region–aptamer pair identified. We used a collinear cut-off of 0.1 and a $p$ value below $5 \times 10^{-8}$ to identify secondary signals in each region. As a quality control step, we fitted a final model including all identified variants for a given genomic region using individual level data in the largest available data set ('Fenland-OMICs') and discarded all variants no longer meeting genome-wide significance.

**Comparison of effect estimates and genomic regions between SomaScan and Olink**. To systematically test for cross-platform consistency of pQTLs, we performed a reciprocal comparison of effect estimates of genome-wide association

analysis of 871 common protein targets using the SomaScan v4 assay ($N = 10{,}708$, $p < 1.004 \times 10^{-11}$) with 12 Olink panels ($N = 485$, $p < 4.5 \times 10^{-11}$, Supplementary Fig. 5) in the Fenland study. This analysis overcomes the biased assessment of previous one-way or within platform replication efforts[4,5,21]. To test the potential influence of sample size on this comparison, we additionally compared the SomaScan-derived pQTLs to published genetic effect estimates for 90 protein targets from the Olink CVD-I panel including up to 22,000 participants from the SCALLOP consortium[8]. We collapsed genetic variants from overlapping protein targets into one signal if they were in strong LD ($r^2 > 0.8$).

We collapsed pQTLs discovered by either platform using a distance-based threshold ($\pm 500$ kB) to define shared ('cross-platform') versus 'platform-specific' pQTLs. This procedure resulted in 479 ($N = 333$ in *cis*, $N = 146$ *trans*, 390 protein targets, Supplementary Data 5) genomic region–protein target combinations for which we had sufficient statistical power to replicate effects, that is, pQTLs observed in the larger SomaScan study that had at least a $p$ value $< 10^{-5}$ when restricting the analysis to the sample of 485 participants with overlapping measurements (see Methods).

We applied the following criteria to consider a pQTL/genomic region to be shared across both platforms: (1) genome-wide significance in either discovery approach of the same SNV or a proxy in high LD ($R^2 > 0.6$) and/or sufficient effect strength to be detected in the smaller Olink sample, and (2) to be directionally concordant (Supplementary Fig. 5). We further performed a regional look-up ($\pm 500$ kB) if the regional sentinels for the SomaScan assay and Olink were not in LD with the respective lead variant and tested if a conditionally independent pQTL in the same region may align (Supplementary Fig. 5).

To facilitate comparison between SomaScan and Olink, we repeated genetic variant–protein target associations within the same sample for which Olink was available. To account for differing sample sizes between the SomaScan data in Fenland and the varying sample sizes within SCALLOP, we recomputed $p$ values by holding the beta estimates constant and re-estimated standard errors using the respective sample size. We considered a predicted $p$ value threshold of $10^{-5}$ to include pQTLs for consistency assessment in case there was evidence for a genome-wide signal from either approach.

**Annotation of pQTLs**. For each identified pQTL we first obtained all SNPs in at least moderate LD ($r^2 > 0.1$) using PLINK (version 2.0) and queried comprehensive annotations using the variant effect predictor software[53] (version 98.3) using the *pick* option. For each *cis*-pQTL we checked whether either the variant itself or a proxy in the encoding gene ($r^2 > 0.6$) is predicted to induce a change in the amino acid sequence of the associated protein, so-called PAVs.

**Phenome-wide association analyses**. To enable linkage to reported GWAS-variants we downloaded all SNPs reported in the GWAS catalogue[54] (19 December 2019) and pruned the list of variant-outcome associations manually to omit previous protein-wide GWASs. For each SNP identified in the present study we tested whether the variant or a proxy in LD ($r^2 > 0.8$) has been reported to be associated with other outcomes previously.

We used the Open GWAS database[55] to query for each genomic region associations with non-proteomic phenotypes using the R package 'ieugwasr' v0.1.5. We tested for a shared genetic signal between a protein target and a phenotype with at least suggestive evidence ($p < 10^{-6}$) using statistical colocalisation[56] as implemented in the R package 'coloc' v4.0.4. We used a conservative prior ($p = 1 \times 10^{-6}$) for the probability of a shared signal between a protein and a trait and further filtered results for protein–phenotype examples for which the respective regional lead variants were in strong LD ($r^2 > 0.8$). We extended this colocalisation approach to all overlapping protein targets with at least suggestive evidence for a *cis*-pQTL for either assay ($p < 10^{-6}$). We considered a posterior probability of 80% as highly likely. We repeated this analysis for all *cis*-regions from the SomaScan-based discovery with evidence for a secondary signal ($p < 5 \times 10^{-8}$) by creating conditional summary statistics using the lead signal in the locus as additional covariate. We computed conditional association statistics using the *cond* option from GCTA-cojo to align with the identification of secondary signals.

**Expression quantitative trait loci**. We obtained lead eQTLs from the most recent release of the GTEx project v8[57] across all 49 tissues and mapped *cis*-pQTLs to *cis*-eQTLs by LD ($r^2 > 0.8$) restricting to the respective protein-encoding gene. We further generated a simple LD-based mapping ($r^2 > 0.8$) considering any overlap between lead pQTLs and eQTLs to allow for incorporation of *trans*-pQTLs.

**Analysis of genetic associations**. To identify factors that are associated with pQTLs that are shared across platforms as opposed to those that are platform-specific, we used logistic regression models to systematically test the odds of platform-specificity for 22 factors, including functional annotation of variants, associations with diverse phenotypic traits, gene eQTL and protein characteristics. We considered three control groups: (1) protein targets with distinct pQTLs in the same genomic region, (2) pQTLs unique to the SomaScan assay and (3) pQTLs unique to the Olink assay (Supplementary Data 7–9). We derived robust standard errors using the sandwich method. We applied log-transformation ('apparent Kd') or square root-

transformation (number of colocalising traits, absolute effect estimate and predicted explained variance) to reduce the impact of highly skewed factors.

To decompose the variance of measurement differences, we computed the differences in rank-transformed measurements between SomaScan and Olink for each overlapping protein target. We used this variable as outcome for a variance decomposition model as implemented in the R package 'variancePartition' v1.14.1 using a corresponding pQTL, age, sex, body mass index, plasma alanine aminotransferase and estimated glomerular filtration rate as explanatory variables. We selected only one pQTL for each overlapping pair based on a simple linear regression model explaining the differences in measurements.

Finally, we used a linear regression model to test whether the association between the Olink measure (outcome) and the SomaScan measure (exposure) differed by genotype of associated pQTLs. The resulting $p$ value for the interaction term between the SomaScan variable and the pQTL can be interpreted as a test of differential correlation coefficients based on the genotype. We accounted for multiple testing by adopting a false discovery rate of 20%. We took a permissive approach given the small sample size ($N = 485$) and the generally low statistical power to detect interaction terms.

We used R version 3.6.0 (R Foundation for statistical computing, Vienna, Austria), including the package 'igraph' v1.2.6, and BioRender.com for visualisation of results.

**Reporting summary**. Further information on research design is available in the Nature Research Reporting Summary linked to this article.

## Data availability
Information about the Fenland cohort is available at the study website (https://www.mrc-epid.cam.ac.uk/research/studies/fenland/information-for-researchers/), which includes a link to the MRC Epidemiology Unit metadata access portal (https://epi-meta.mrc-epid.cam.ac.uk/). To comply with the consent given by Fenland participants, data access is granted to bona fide researchers through an application process that typically takes no more than 4–6 weeks. Data will either be shared through an institutional data sharing agreement or arrangements will be made for analyses to be conducted remotely without the necessity of data transfer. Publicly available summary statistics for look-up and colocalisation of pQTLs were obtained from https://gwas.mrcieu.ac.uk/ and https://www.ebi.ac.uk/gwas/. We obtained genome-wide summary statistics for 90 protein targets from Folkersen et al.[8], which are also available from the GWAS catalogue (https://www.ebi.ac.uk/gwas/publications/33067605, GCST90011994-GCST90012083). The Cryo-EM structure for GDF-15 and associated receptors has been obtained from the Protein Data Bank 6Q2J. Source Data are provided with this paper.

## Code availability
Code used in the present study has been deposited on GitHub at https://github.com/MRC-Epid/cross_platform_pGWAS.

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

## Acknowledgements

The Fenland Study (10.22025/2017.10.101.00001) is funded by the Medical Research Council (MC_UU_12015/1). We are grateful to all the volunteers and to the General Practitioners and practice staff for assistance with recruitment. We thank the Fenland Study Investigators, Fenland Study Co-ordination team and the Epidemiology Field, Data and Laboratory teams. We further acknowledge support for genomics from the Medical Research Council (MC_PC_13046). Proteomic measurements were supported and governed by a collaboration agreement between the University of Cambridge and Somalogic. J.C.-Z. is supported by a 4-year Wellcome Trust PhD Studentship and the Cambridge Trust. M.K. is supported by a Gates Fellowship. C.L., M.P., E.W., J.L., E.O., N.D.K., and N.J.W. are funded by the Medical Research Council (MC_UU_12015/1). N.J.W. is a NIHR Senior Investigator. A.D.H. is an NIHR Senior Investigator and supported by the UCL Hospitals NIHR Biomedical Research Centre and the UCL BHF Research Accelerator (AA/18/6/34223). We thank Philippa Pettingill, Ida Grundberg, Klev Diamanti and Andrea Ballagi for advice and comments on an earlier draft of this manuscript. We thank Vladimir Saudek for generating a 3D model of variant GDF-15 protein. This work was supported in part by the UKRI/NIHR Strategic Priorities Award in Multimorbidity Research for the Multimorbidity Mechanism and Therapeutics Research Collaborative (MR/V033867/1).

## Author contributions

M.P. and C.L. designed the analysis and drafted the manuscript. M.P., E.W. and J.L. analysed the data. J.C.-Z. and E.O. provided bioinformatic characterisation of protein targets and mapped pQTLs to eQTLs. N.D.K. and E.O. performed quality control of proteomic measurements. A.D.H. and S.A.W. provided critical review and intellectual contribution to the discussion of results. N.J.W. is PI of the Fenland cohort. All authors contributed to the interpretation of the results and critically reviewed the manuscript.

## Competing interests

S.A.W. is an employee of SomaLogic. The remaining authors declare no competing interests.
