## [Peer Review File · Nature Communications]

REVIEWER COMMENTS

Reviewer #1 (Remarks to the Author):

Pietzner et al. compare the performance and output of two affinity-based proteomics technologies, the aptamer-based SOMAscan and the Olink assay, using 871 overlapping protein targets (measured in plasma of 485 individuals). The comparison included the correlation between protein targets as well as the identification of protein quantitative trait loci (pQTLs) and their associations with a variety of phenotypes based on a range of technical factors that may affect these platforms differently. In this case, the correlation was modest (median $r = 0.38$), more than 60% of the pQTLs are shared (directionally consistent) by the two platforms, and roughly 36% of protein-to-trait associations were specific for one platform.

The authors argue that their findings show how these two platforms can be used in tandem to better understand and identify novel pathobiology of disease, and that they can serve as a paradigm for future cross-platform discoveries. This is a reasonable conclusion, but it does not provide a scientific novelty in and of itself, emphasizing that this study is primarily a technical report rather than offering new scientific discoveries. This is not intended to diminish the work's conclusions which are valuable, but rather to raise the question of whether a descriptive study of this kind is suitable for the journal. My other major reservations about the manuscript are as follows:

1. On page 5 (lines 92-100), the authors discuss the number of aptamer-to-trait associations as an indicator of whether or not a normalization step is used. First, there is no mention of how the normalization was performed or cited, nor is there any explanation for why the number of aptamer-to-trait associations is a good estimate for using the normalization step. Similarly, are such normalization procedures not required for the Olink assay? Finally, the wording is a little off, such as the use of disproportional in this context.

2. A similar median correlation (median $r = 0.36$) between proteins targeted by these two platforms was previously observed in a small sample by Raffield et al. (PMID: 32386347). However, they discovered that aptamer specificity as assessed by orthogonal measures (mass spectrometry) in another study was enriched in the high correlation group. The author should consider whether this holds true in their much larger context.

3. On page 6 (lines 132-133): "We observed a lower fraction of shared genomic regions when comparing to publicly available Olink pQTLs with 39.1%, which was best explained by the presence

of multiple non-specific trans-pQTLs (see Supplementary Note and Supplementary Tab. 5).” It is difficult to see what supports this statement from either the Suppl note or the Suppl table S5.

4. The effect of various factors on the platform specificity of the identified pQTLs is depicted in Figure 3c. I would expect that these various factors influence cis and trans effects in different ways, resulting in different outcomes. As a result, the authors should present these results (i.e. platform specificity) separately for cis and trans pQTLs.

5. On page 7 (lines 144-148) “In other words, the agreement between both platforms was higher for a genetically defined subgroup of 146 participants, with effects in cis possibly pointing to epitope effects, whereas effects in trans pointing towards posttranslational modifications, such as glycosylation (Supplementary Tab. 9 and Supplementary Note)”. There is nothing in either the Suppl note or the Suppl table 9 that supports this. At this point, it appears to be a pure conjecture.

6. This study requires a much more detailed comparison of the differences in detection sensitivity and dynamic range offered by these two different platforms.

7. The Olink platform's measurement of a smaller number of proteins accounts for the relatively low overlap of 871 proteins between these two platforms. The question of what distinguishes these protein targets from the other proteins measured by the SOMAscan panel remains unanswered. Are overlapping proteins, for example, more likely to be secreted? Are they more abundant? Is it more likely that the overlapping aptamers will hit the specific target? Is it likely that the overall comparison would hold if the overlap (i.e. Olink measured more proteins) was greater (for discussion)?

8. The effect of different proteoforms (due to structural variants) on outcome is mentioned throughout the paper (GDF15 is an example), and this may explain the association to a phenotype to some extent. It's difficult to see how this differs from artifactual effects on altered epitope binding. The authors should be more specific and clarify what they mean by the effect of different proteoforms on outcome. More specifically, how can altered binding affinity of aptamers detect differences in activity rather than abundances?

9. In Supplementary note (lines 26-31): “Firstly, variants in trans might increase DNA-binding affinity of abundant circulating proteins such as complement factor H (rs1061170 within CFH) or Butyrylcholinesterase (rs1803274 within BCHE) possibly interfering with SOMAmer reagents¹⁹, and, secondly, reflect study- specific handling of blood samples like rs3443671 within NLRP12, which might only be identified as a pQTL as a result of white blood cell lysis. Out of the 140 platform-

specific trans-pQTLs, 26 and 25, respectively, were likely attributable to those reasons." Reference 19, for example, does not mention BCHE, and there is no mention of the effect at the NLRP12 locus. Is it possible to find a list of the 51 pQTLs that the authors believe affect DNA binding or sample handling?, or are there any definitions or experimental data to back up these assertions?

Reviewer #2 (Remarks to the Author):

In this manuscript, Pietzner and collaborators report a systematic comparison of two of the most popular platforms for affinity-based proteomics. They describe findings regarding shared and platform-specific protein biology and look deeper into a handful of examples. They conclude that these two platforms are "synergistic". I provide below recommendations to make their conclusions more reliable.

1. A significant portion of the results for the paper are based on comparing results across a subset of individuals (485). I couldn't find anywhere on the manuscript what was the inclusion/exclusion criteria to select these individuals. Authors should report in a table a detailed comparison of these 485 subjects to the rest of the study with regards to key parameters that could influence the interpretation of the results (i.e., age, gender, ethnicity, technical variables, any difference in genotyping platform, etc.).

2. Figure 2B, Supplementary Note. It is evident from Figure 2B that a very large proportion of the variation between the platforms is accounted for the % of outliers in the SomaScan platform. The authors don't mention this on the main text and barely mention it on the Supplementary Note. If this is the most important difference, the authors should explore with more detail if this is a technical artifact, or if the SomaScan platform is able to truly detect those real outliers and report it. Is this observation consistent across all measured proteins? is it most obvious in a particular protein family?

3. Page 6, line 126. " This included 13 regions for which we discovered two independent cis-pQTLs ($R^2 < 0.1$) for SomaScan but only the secondary signal was shared with Olink." Please provide regional plots for these 13 loci highlighting the secondary signals.

4. Page 6 and 7, lines 140-147. The authors claim that some of the differences observed between the platforms are genotype specific. It would be useful for the wide audience to provide an example of what exactly they mean. For example, the most remarkable finding is with rs2071579. That variant is in almost complete LD ($r^2=0.997$) with a missense variant rs880633. It would be useful if the authors could go into details of this observation and see if the optamer has better binding to the alternative aminoacid that could explain this finding.

5. Pages 9-10, lines 192-205. The discordant direction of effects observed for PILRA and AD are of potential relevance and I feel this should be one of the key highlighted results of this paper (perhaps even in the abstract). It clearly shows how one platform can give you completely opposite interpretations to the other. The interpretation of the direction of effects has important implications for drug development. I'd advise to provide a new figure that captures these findings in a similar schematic way of Figure 6.

6. Page 11. Lines 242: " The variant is in strong LD ($r^2=0.99$) with rs9427397, whose T-allele (allele frequency=14.1%) introduces a premature stop codon possibly".

This interpretation is wrong, it was curated by gnomad researchers as a multinucleotide variant (MNV), This variant is in phase with 1-161476205-A-G, altering the amino acid sequence, so instead of creating a stop codon individuals get a missense change (W). For more details look at: https://gnomad.broadinstitute.org/variant/1-161476204-CA-TG?dataset=gnomad_r2_1

Minor.

Page 11. line 295. Add a supplementary figure of the 3D model of GDF-15 with p.H202D or p.H6D

Reviewer #3 (Remarks to the Author):

The present paper describes a comparison between two affinity-based proteome profiling for the identification of protein quantitative trait loci (pQTLs) in a large British population-based cohort. A total of 1,923 pQTLs have been identified for the 871 overlapping proteins. A gene-protein-phenotype network has also been constructed based on colocalisation analysis with the integration

of publicly available GWAS data. The results shown in the study are indeed helpful since both approaches have been widely used in clinical samples. Overall, the study is well-conducted with appropriate statistics and workflow. The manuscript is also well structured and well written.

Still, some specific unclear points that the authors should address:

- 1) What is the inclusion criteria of the Fenland study? The procedure of sample selection for Olink should be also provided since all the samples are from the same test site.
- 2) The normalization procedure for SomaLogic data is missing as well as the batch information. The authors should show details on how the AMN dataset was generated.
- 3) The technical factors for each protein target in each platform can be included as a supplementary table. From Figure 2b, proteins with most outliers or under LOD values are the ones with poor correlations cross platform. But there seems to be no filtering process on protein level for quality control.
- 4) I try to check the list of platform-specific pQTLs, but the information in Table S2 is very confusing. For example, rs73224660 - BST1 is supposed to be a SOMAScan specific pQTL, but the pvalue in Olink is also very low (2.44×10^{-46}) and the same genetic region has been published before in Olink pQTL studies (Suhre et al. Nature Communications, 2017 ; Zhong et al. Nature Communications, 2021).

We have now revised our article in light of the three reviewers' suggestions.

As suggested, we provide more detailed assessment of technical factors and have clarified hypothesis and explanations for inconsistencies observed across platforms, in particular with respect to pQTLs. You will see that most of the suggested changes referred to technicalities best presented in the Supplemental Note to not further expand the technical part of the paper as recommended by the reviewers. We have further revised key figures presenting our findings with the clear aim to highlight biological insights drawn from our data. For example, Figure 5 contains now schematics on how protein altering variants that might have otherwise been considered as technical artefacts provide unique insights on possible downstream consequences of proteoforms.

Please find a detailed point-by-point response to the comments of the reviewers below.

REVIEWER COMMENTS

Reviewer #1 (Remarks to the Author):

Pietzner et al. compare the performance and output of two affinity-based proteomics technologies, the aptamer-based SOMAscan and the Olink assay, using 871 overlapping protein targets (measured in plasma of 485 individuals). The comparison included the correlation between protein targets as well as the identification of protein quantitative trait loci (pQTLs) and their associations with a variety of phenotypes based on a range of technical factors that may affect these platforms differently. In this case, the correlation was modest (median $r = 0.38$), more than 60% of the pQTLs are shared (directionally consistent) by the two platforms, and roughly 36% of protein-to-trait associations were specific for one platform.

The authors argue that their findings show how these two platforms can be used in tandem to better understand and identify novel pathobiology of disease, and that they can serve as a paradigm for future cross-platform discoveries. This is a reasonable conclusion, but it does not provide a scientific novelty in and of itself, emphasizing that this study is primarily a technical report rather than offering new scientific discoveries. This is not intended to diminish the work's conclusions which are valuable, but rather to raise the question of whether a descriptive study of this kind is suitable for the journal. My other major reservations about the manuscript are as follows:

1. On page 5 (lines 92-100), the authors discuss the number of aptamer-to-trait associations as an indicator of whether or not a normalization step is used. First, there is no mention of how the normalization was performed or cited, nor is there any explanation for why the number of aptamer-to-trait associations is a good estimate for using the normalization step. Similarly, are such normalization procedures not required for the Olink assay? Finally, the wording is a little off, such as the use of disproportional in this context.

***R1 response 1** We added the missing information about the AMN procedure to method section of the revised manuscript (page 13, lines 365-370). The AMN procedure ensures transferability of protein models to independent cohorts, which wouldn't otherwise be possible using semiquantitative measurements.*

We agree that the number of aptamer-to-trait associations is not necessarily a good measure to decide whether a normalization step should be performed. We aimed to be transparent about possible downstream consequences of major technical differences between the platforms and have rephrased the section accordingly.

Due to its panel-based nature, no such normalisation step is recommended by Olink.

2. A similar median correlation (median $r = 0.36$) between proteins targeted by these two platforms was previously observed in a small sample by Raffield et al. (PMID: 32386347). However, they discovered that aptamer specificity as assessed by orthogonal measures (mass spectrometry) in another study was enriched in the high correlation group. The author should consider whether this holds true in their much larger context.

R1 response 2 *Thank you for this suggestion; we have now performed a similar analysis as Raffield et al. and also observed on average higher correlation coefficients (median correlation: 0.57 vs 0.27, p -value < 1.59×10^{-21}) for protein targets with existing orthogonal validation, including by mass spectrometry analysis of pull-down assays or measurements using different assays. We have now added this information to the revised version of the manuscript (p4, line 94-98).*

3. On page 6 (lines 132-133): “We observed a lower fraction of shared genomic regions when comparing to publicly available Olink pQTLs with 39.1%, which was best explained by the presence of multiple non-specific trans-pQTLs (see Supplementary Note and Supplementary Tab. 5).” It is difficult to see what supports this statement from either the Suppl note or the Suppl table S5.

R1 response 3 *We have rephrased this section in the Supplemental Note for clarity (p2, line 42-49). The major difference between the two pQTL sets, Fenland and Folkersen et al. (publicly available Olink pQTLs), is the fraction of cis-to-trans pQTLs. Folkersen et al. focused on a specific set of 90 proteins related to cardiovascular health, with comparably few cis-pQTLs ($n=67$, 55.2% replication), but many trans-pQTLs ($n=240$, 34.6% replication). As outlined more in detail our response to comment 9 below, a substantial fraction of trans-pQTLs not replicating is due to platform-specific effects of the SomaScan assay and pleiotropic but unspecific trans-pQTLs in Folkersen et al.*

The Supplemental Note now reads as follows:

“Firstly, variants in trans might increase DNA-binding affinity of abundant circulating proteins such as complement factor H^2 (rs1061170 within CFH) or alter the activity of enzymes with an affinity to a large spectrum of chemical entities such as butyrylcholinesterase (rs1803274 within BCHE known to reduce enzymatic activity³) thereby possibly interfering with SOMAmer reagents. Secondly, samples taking from participants with a higher genetic susceptibility to (white) blood cell counts are possibly more prone to analytical artefacts during sample preparation, such as cell lysis and subsequent spill over of proteins into the plasma. The pleiotropic trans-pQTL rs3443671 within NLRP12 might be such an example, since neither we or other SomaScan-based discovery efforts⁴ were able to replicate this pleiotropic association, which is a known blood cell locus⁵.”

4. The effect of various factors on the platform specificity of the identified pQTLs is depicted in Figure 3c. I would expect that these various factors influence cis and trans effects in different ways, resulting in different outcomes. As a result, the authors should present these results (i.e. platform specificity) separately for cis and trans pQTLs.

R1 response 4 *We agree with the reviewer that stratified or even better effect modification analysis by cis/trans status would provide further insights into platform (in)consistencies. However, the extreme imbalance of cis to trans pQTLs (distinct pQTLs ($N=cis/trans$): $N=94/2$; unique to Olink: $N=49/10$; unique to SomaScan: $N=17/42$) and hence the low number of outcomes in stratified analysis*

did not permit any such analysis and we added a statement to the limitation section of the manuscript to acknowledge possible cis/trans-specific effects (p12, line 334-336).

5. On page 7 (lines 144-148) “In other words, the agreement between both platforms was higher for a genetically defined subgroup of participants, with effects in cis possibly pointing to epitope effects, whereas effects in trans pointing towards posttranslational modifications, such as glycosylation (Supplementary Tab. 9 and Supplementary Note)”. There is nothing in either the Suppl note or the Suppl table 9 that supports this. At this point, it appears to be a pure conjecture.

***R1 response 5** We agree with the reviewer that our previous explanations missed the level of detail to support the claim made in the main text. The updated Supplementary Note now provides a more detailed explanation (p3, line63-66). Briefly, strong genetic effects in cis seen only for either assay might likely point towards interference with the measurement technique, most likely due to a different shape of the target protein through a common missense variant. In this case, measurements among homozygous carriers of the non-effect allele are likely to be unaffected and hence align better with the other assay. A similar explanation might hold true for some trans-pQTLs, whereby the trans-pQTL may alter the activity of the enzyme encoded nearby, such as glycosyltransferases like the histo-blood group ABO system transferase, which in turn interacts with the protein target leading to, for instance, altered glycosylation patterns and hence accessibility of the protein target to affinity reagents.*

6. This study requires a much more detailed comparison of the differences in detection sensitivity and dynamic range offered by these two different platforms.

***R1 response 6** We agree and while we do not have the information required to assess and compare specific characteristics of the assay performance for each technology, we have now added all information we had available (i.e. non-proprietary information) regarding these aspects, including the number of values below LOD (as a proxy for detection sensitivity), binding affinity of SOMAmer reagents, dilution bin (as a proxy for the dynamic range), as well as derived QC measures, including the percentage of outliers, to Supplementary table 2.*

7. The Olink platform's measurement of a smaller number of proteins accounts for the relatively low overlap of 871 proteins between these two platforms. The question of what distinguishes these protein targets from the other proteins measured by the SOMAscan panel remains unanswered. Are overlapping proteins, for example, more likely to be secreted? Are they more abundant? Is it more likely that the overlapping aptamers will hit the specific target? Is it likely that the overall comparison would hold if the overlap (i.e. Olink measured more proteins) was greater (for discussion)?

***R1 response 7** In response to this helpful recommendation, we have now systematically tested for differences in proteins characteristics between those proteins that do versus do not overlap. We found an enrichment of secreted proteins (odds ratio 3.66, p-value<4.7e-44) and high-affinity targets (odds ratio 1.18, p-value<4.3e-6), whereas proteins with a high percentage of outlying values were slightly depleted (odds ratio 0.87, p-value<1.3e-4). Since these results indicate a possible selection bias towards well-correlated protein targets, we added a cautionary note to the main text (p14, line 400-404).*

8. The effect of different proteoforms (due to structural variants) on outcome is mentioned throughout the paper (GDF15 is an example), and this may explain the association to a phenotype to some extent. It's difficult to see how this differs from artefactual effects on altered epitope binding. The authors should be more specific and clarify what they mean by the effect of different proteoforms on outcome. More specifically, how can altered binding affinity of aptamers detect differences in activity rather than abundances?

R1 response 8 We followed the recommendation of the reviewer and have now added a dedicated section to the discussion explaining the concept of ‘proteoforms’ in more detail (p11, line 298-308). We agree with the reviewer and now clarify that missense variation or splicing QTLs giving rise to alternative proteoforms are indistinguishable from measurement artefacts, and that it is only by triangulating with phenotypic follow-up that inference on protein function rather than protein abundances might be hypothesised. Consider for instance the PILRA example, in which a common missense variant, detected by both assays, confers a lower risk for Alzheimer’s disease by diminishing the capacity of PILRA to bind ligands. To mitigate speculation, we carefully selected examples for which both assays had distinct lead cis-pQTLs for the same protein target, ensuring that at least on of the two instruments protein abundance, whereas the other one relates to the shape of the target protein.

In general, testing for a shared genetic signal between protein measurements and phenotypes enhances confidence of the candidate causal gene and further provides evidence in vivo that the variant protein is indeed expressed and detectable at physiological concentrations in the circulation. While some of those cis-pQTLs linked to PAVs may indeed represent pure measurement artefacts with no obvious downstream consequences, linkage of, often benign, PAVs to diseases via the proteome is important to understand the role of proteins in diseases.

9. In Supplementary note (lines 26-31): “Firstly, variants in trans might increase DNA-binding affinity of abundant circulating proteins such as complement factor H (rs1061170 within CFH) or Butyrylcholinesterase (rs1803274 within BCHE) possibly interfering with SOMAmer reagents¹⁹, and, secondly, reflect study- specific handling of blood samples like rs3443671 within NLRP12, which might only be identified as a pQTL as a result of white blood cell lysis. Out of the 140 platform-specific trans-pQTLs, 26 and 25, respectively, were likely attributable to those reasons.” Reference 19, for example, does not mention BCHE, and there is no mention of the effect at the NLRP12 locus. Is it possible to find a list of the 51 pQTLs that the authors believe affect DNA binding or sample handling?, or are there any definitions or experimental data to back up these assertions?

R1 response 9 We apologize for the misleading reference and have updated the Supplemental Note accordingly (p2, line 40-49) and included a flag in Supplementary Table 5. Briefly, we observed multiple pleiotropic trans-pQTLs that were specific to either the SomaScan-based discovery or the publicly available Olink pQTLs. We observed two possible mechanisms explaining such: 1) genetic susceptibility to higher (white) blood cell counts (as for NLRP12), and 2) increased DNA-binding capacity possibly conferred by functional variants in LD (as for CFH).

We have previously hypothesised that the NLRP12 locus (Pietzner et al. 2020 Nature Communications) is a study-specific artefact, in line with other SomaScan studies not replicating this locus (e.g., Gudjonsson et al. 2021 BioRxiv). A possible explanation being, that a genetically higher white blood cell count makes whole blood samples more vulnerable to sampling artefacts induced by cell lysis. A possible interference of the SomaScan technology with DNA-binding proteins has already been suggested (Suhre et al. 2020 Nat Rev Gen) and experimental evidence showed an increased DNA-binding capacity of complement factor H (encoded by CFH) conferred by the missense variant rs1061170 (Sjörberg et al. 2007 Biol. Chem).

Given the similar unspecific pleiotropic nature of rs1803274 (p.A567Thr) within BCHE (associated with >500 aptamers in the entire data set), we speculated that this might represent a measurement artefact rather than a truly pleiotropic effect on the proteome. The variant, also called K-variant, is well described for lowering the activity of butyrylcholinesterase, an abundant but otherwise purely characterized enzyme, that is secreted into blood.

Reviewer #2 (Remarks to the Author):

In this manuscript, Pietzner and collaborators report a systematic comparison of two of the most popular platforms for affinity-based proteomics. They describe findings regarding shared and platform-specific protein biology and look deeper into a handful of examples. They conclude that these two platforms are "synergistic". I provide below recommendations to make their conclusions more reliable.

1. A significant portion of the results for the paper are based on comparing results across a subset of individuals (485). I couldn't find anywhere on the manuscript what was the inclusion/exclusion criteria to select these individuals. Authors should report in a table a detailed comparison of these 485 subjects to the rest of the study with regards to key parameters that could influence the interpretation of the results (i.e., age, gender, ethnicity, technical variables, any difference in genotyping platform, etc.).

R2 response 1 Apologies for this omission, we have now added this information, including a table describing potential differences between the entire Fenland cohort and the subcohort selected for proteomic profiling with Olink (now Supplemental Table 1). In brief, participants were selected at random from all Fenland participants that were eligible, based on being recruited at the one of the three study centres and genotyped using the same array for pragmatic reasons. Included individuals were broadly representative of the total sample, as indicated by a similar age, BMI and sex ratio.

2. Figure 2B, Supplementary Note. It is evident from Figure 2B that a very large proportion of the variation between the platforms is accounted for the % of outliers in the SomaScan platform. The authors don't mention this on the main text and barely mention it on the Supplementary Note. If this is the most important difference, the authors should explore with more detail if this is a technical artifact, or if the SomaScan platform is able to truly detect those real outliers and report it. Is this observation consistent across all measured proteins? is it most obvious in a particular protein family?

R2 response 2 In response to this comment, we have now performed additional analyses to identify possible factors contributing to a high number of outliers and found that the presence of a cis-pQTL for the SomaScan assay (inversely, $p < 3.1e-23$), dilution bin (highest in the undiluted bin, i.e., least abundant proteins, $p < 4.7e-12$), binding affinity of the SomaScan reagent (inversely, $p < 4.1e-10$), and Olink panel (highest for the inflammatory panel, $p < 6.5e-10$) were associated factors. Some of these factors point to the relevance of measurement artefacts, such as lower binding affinity of the SOMAmer reagent or a lower likelihood of orthogonal validation of the protein target. However, closer inspection of proteins with very higher values of %-outliers, revealed consistent results between SomaScan and Olink, and therefore possibly true outlying groups. For example, the proteins Cripto and FOLR3 had high fractions of samples off the median, which were explained by strong and consistent effects of cis-pQTLs (see Figure 1 in the revised Supplemental Note). We added this information to the Supplemental Note (p1, line 13-21).

3. Page 6, line 126. " This included 13 regions for which we discovered two independent cis-pQTLs ($R^2 < 0.1$) for SomaScan but only the secondary signal was shared with Olink." Please provide regional plots for these 13 loci highlighting the secondary signals.

R1 response 3 We have added the requested plots to the Supplemental Material (Supplementary Figure 6 and 7).

4. Page 6 and 7, lines 140-147. The authors claim that some of the differences observed between the platforms are genotype specific. It would be useful for the wide audience to provide an example of what exactly they mean. For example, the most remarkable finding is with rs2071579. That variant is in almost complete LD ($r^2=0.997$) with a missense variant rs880633. It would be useful if the authors could go into details of this observation and see if the aptamer has better binding to the alternative aminoacid that could explain this finding.

R2 response 4 We followed these helpful suggestions and extended the scheme presenting the possible impact of genetic variants on affinity-based proteomic measurements to the revised Figure 6. We further elaborate on the possible impact of common missense variation (rs880633 has a MAF of 47.0%) within protein substructures that are predicted to be readily accessible for antibody binding (the arginine to glycine substitution, p.R145G, introduced by rs880633 falls into such a region) on binding affinity using chitinase-3-like protein 1 as an example (p6, line 131-138).

We note that while the correlation shown in Figure 3 is highest for carriers of the minor allele, the amino acid substitution introduced by the missense variant is conferred by the major allele, and hence both assays agree best for carriers of the minor but 'wild-type' allele.

5. Pages 9-10, lines 192-205. The discordant direction of effects observed for PILRA and AD are of potential relevance and I feel this should be one of the key highlighted results of this paper (perhaps even in the abstract). It clearly shows how one platform can give you completely opposite interpretations to the other. The interpretation of the direction of effects has important implications for drug development. I'd advise to provide a new figure that captures these findings in a similar schematic way of Figure 6.

R2 response 5 In response to this comment we have now created a new panel figure to present this finding in the main text (now Figure 5a) and included a reference to this example in the abstract.

6. Page 11. Lines 242: " The variant is in strong LD ($r^2=0.99$) with rs9427397, whose T-allele (allele frequency=14.1%) introduces a premature stop codon possibly".

This interpretation is wrong, it was curated by gnomad researchers as a multinucleotide variant (MNV), This variant is in phase with 1-161476205-A-G, altering the amino acid sequence, so instead of creating a stop codon individuals get a missense change (W). For more details look at: https://gnomad.broadinstitute.org/variant/1-161476204-CA-TG?dataset=gnomad_r2_1

R2 response 6 Thank you for pointing out our error! We have revised the corresponding section accordingly (p8, line 205-208).

Minor.

Page 11. line 295. Add a supplementary figure of the 3D model of GDF-15 with p.H202D or p.H6D

R2 response 7 We have now included a new Supplemental figure 10 that shows a 3D-model which highlighting p.H6D in a complex of GDF-15 dimer and its currently sole known receptor GFRAL, demonstrating why the variant is currently considered benign.

Reviewer #3 (Remarks to the Author):

The present paper describes a comparison between two affinity-based proteome profiling for the identification of protein quantitative trait loci (pQTLs) in a large British population-based cohort. A total of 1,923 pQTLs have been identified for the 871 overlapping proteins. A gene-protein-phenotype network has also been constructed based on colocalisation analysis with the integration of publicly available GWAS data. The results shown in the study are indeed helpful since both approaches have been widely used in clinical samples. Overall, the study is well-conducted with appropriate statistics and workflow. The manuscript is also well structured and well written. Still, some specific unclear points that the authors should address:

We thank the reviewer for his/her kind summary of our work.

1) What is the inclusion criteria of the Fenland study? The procedure of sample selection for Olink should be also provided since all the samples are from the same test site.

R3 response 1 We have now included a more detailed description of sample selection (p13, line 385-389) as well as study characteristics for the Olink sub-cohort (now Supplementary Table S1) in the revised version of the manuscript. Please also see R2 response 1.

2) The normalization procedure for SomaLogic data is missing as well as the batch information. The authors should show details on how the AMN dataset was generated.

R3 response 2 We have now included the suggested information in the revised version of the manuscript (p13, line 385-388).

3) The technical factors for each protein target in each platform can be included as a supplementary table. From Figure 2b, proteins with most outliers or under LOD values are the ones with poor correlations cross platform. But there seems to be no filtering process on protein level for quality control.

R3 response 3 As requested, we have now included this information in Supplementary Table 2. To provide some context regarding the lack of filtering: this was deliberately not done in order to be able to investigate the influence of the metrics potentially indicating poor measurement performance (such as high % below LOD) on observational correlations and consistency of genetic signals, one of the aims of this study. Excluding proteins on this basis would have likely led to biased estimation of the consistency of genetic signals. Also, in general, QC metrics provided by each vendor were excellent for the majority of proteins, but this does not necessarily equate to consistency across platforms; in other words, even good QC parameters may indicate good measurements but in the context of poor cross-platform correlation of a different protein target or isoform. To illustrate or uncover these separate points, which we think are useful for the readership of Nature Communications, we included as many proteins as possible without selection. However, we of course completely agree with the reviewer, that now that these principles have been established, such filtering can be a useful step to prioritise pQTLs more likely to be consistent across platforms even in the absence of data on multiple platforms in a single study.

4) I try to check the list of platform-specific pQTLs, but the information in Table S2 is very confusing. For example, rs73224660 - BST1 is supposed to be a SOMAScan specific pQTL, but the pvalue in Olink is also very low (2.44×10^{-46}) and the same genetic region has been published before in Olink pQTL studies (Suhre et al. Nature Communications, 2017 ; Zhong et al. Nature Communications, 2021).

R3 response 4 We agree that this was not clear enough and have now added a more detailed explanation what the last column ('discovery') refers to. In brief, we discovered two distinct variants associated with BST1 as measured by SomaScan (rs73224660) and Olink (rs55735476). These variants were in strong LD ($r^2=0.77$) but below our threshold of 0.8 used in our study for collapsing signals. In a detailed and more rigorous regional comparison, both signals align and were flagged as platform consistent (Supplementary Table 5). We chose such a two-staged approach to 1) be consistent what previous efforts have done, and 2) to contrast the overall consistency, that is across all proteins and SNPs, with a more rigorous assessment of the local genetic architecture.

REVIEWERS' COMMENTS

Reviewer #1 (Remarks to the Author):

In responding to my comments/suggestions, the authors have done an excellent job. Furthermore, I appreciate the new Fig. 5 created in response to reviewer 2's comment.

Reviewer #2 (Remarks to the Author):

All my comments have been addressed, I recommend accepting with the publication of this paper.

Reviewer #3 (Remarks to the Author):

The authors have addressed my concerns with additional data. I agree that the paper has been improved. Therefore, I suggest this manuscript is suitable to be accepted in Nature Communications.